# GENERALIZABLE LLM LEARNING OF GRAPH SYNTHETIC DATA WITH POST-TRAINING ALIGNMENT

## ABSTRACT

Previous research has sought to enhance the graph reasoning capabilities of LLMs by supervised fine-tuning on synthetic graph data. While these led to specialized LLMs better at solving graph algorithm problems, we don't need LLMs for shortest path: we need generalization from synthetic graph data to real-world tasks with implicit graph structures. In this work, we propose to unlock generalizable learning of graph with post-training alignment with synthetic data. We first design *solution-based* and *process-based* rewards for synthetic graph problems: instead of rigid memorizing response patterns in direct fine-tuning, we posit that post-training alignment would help LLMs grasp the essentials underlying graph reasoning and alleviate overfitting on synthetic data. We employ post-training alignment algorithms such as GRPO and DPO, aligning both off-the-shelf LLMs and LLMs fine-tuned on synthetic graph data. We then compare them against existing settings on both in-domain synthetic tasks and out-of-domain real-world tasks with implicit graph structures such as multi-hop QA, structured planning, and more. Extensive experiments demonstrate that our post-training alignment recipe leads to statistically significant improvement on 5 datasets, with an average gain of 12.9% over baseline settings. Further analysis reveals that process-based rewards consistently outperform solution-based rewards on synthetic data but not on real-world tasks, and compositionality and explainable intermediate steps remains a critical challenge even after post-training alignment. [1]

## 1 INTRODUCTION

Going beyond language processing tasks, large language models (LLMs) are increasingly adopted for tasks with implicit graphical structures, such as commonsense reasoning (Sakaguchi et al., 2021; Saha et al., 2021), multi-hop QA (Ho et al., 2020; Ding et al., 2024; Geva et al., 2021), and planning (Valmeekam et al., 2023; Padmakumar et al., 2022). Existing research focuses on evaluating LLMs on the underlying graph problems such as connectivity and shortest path, revealing that LLMs *do* have preliminary graph reasoning capabilities (Wang et al., 2023) while suffering from limitations such as robustness and hallucination (Wang et al., 2023; Zhang et al., 2024b; Guo et al., 2023). Recent works seek to further improve LLM graph reasoning, mostly through supervised fine-tuning (SFT) on *graph synthetic data* (He et al., 2024b; Perozzi et al., 2024): these techniques demonstrate substantial improvement in LLMs' ability to tackle graph algorithm problems.

However, we don't need LLMs to solve synthetic graph problems such as shortest path and topological sort: we already have algorithms (e.g., Dijkstra and Ford-Fulkerson) that are 100% accurate and much more efficient than calling an LLM. The goal of leveraging synthetic graph data should thus be learning from the structured reasoning data and *generalizing from synthetic graph problems to real-world tasks* with implicit structures. Unfortunately, existing SFT recipes are not improving real-world performance (Zhang et al., 2024b), even leading to a 12.5% drop for tasks such as Proscript (Sakaguchi et al., 2021).

To this end, we propose to unlock the power of synthetic graph data with post-training alignment. Instead of directly memorizing reasoning chains of synthetic problems through SFT, we posit that

---

[1]Experimental code and results are available at `https://anonymous.4open.science/r/Graph_RL-BF08/readme.md`

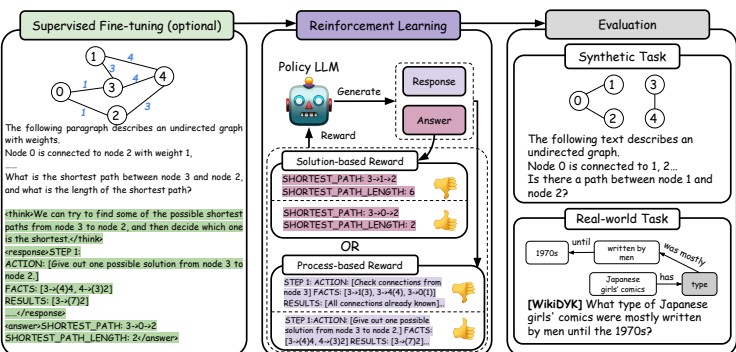

Figure 1: The overall pipeline of our work. We first perform an optional SFT stage (left), followed by RL stage (middle) with two rule-based reward designs. Both SFT and RL training use purely synthetic graph tasks. At last, we evaluate models on both synthetic and real-world tasks (right).

alignment would help LLMs learn the underlying objectives of graph reasoning and mitigate excessive over-fitting to synthetic problems. We design two types of rewards for synthetic graph problems: *solution-based*, where only the final answer is considered; *process-based*, where varying weights are given to the soundness of intermediate steps and the final answer. We then align LLMs, both off-the-shelf and fine-tuned on synthetic graph problems, with alignment algorithms (e.g., GRPO (Shao et al., 2024) and DPO (Rafailov et al., 2023)) using these reward models. We then compare these alignment models against existing settings (off-the-shelf models, SFT, etc.) on 1) in-domain synthetic tasks such as connectivity and shortest path; and more importantly, 2) out-of-domain real-world graph problems such as multi-hop QA (Geva et al., 2021; Ding et al., 2024) and structured commonsense reasoning Saha et al. (2021); Sakaguchi et al. (2021).

Extensive experiments with two LLMs and 8 task settings demonstrate that our proposed alignment recipes can sometimes outperform baselines across tasks and models on both synthetic and real-world graph problems, with important findings on reasoning on gaps between single-step and multi-step reasoning. For synthetic problems, post-training aligned models improve by 25% on average for the connectivity task. More importantly, for real-world graph problems, our recipe provides statistically significant improvements on 5 task settings, with an average improvement of 13%. Our experiments also offer insights into the alignment recipe: the on-policy RL algorithm GRPO outperforms DPO by 5% on average; process-based rewards offer finer-grained signals for models to learn from when using GRPO on synthetic tasks, outperforming solution-based by 24% on average, while there is no significant edge of process-based synthetic rewards on real-world tasks. Further analysis reveals that the main bottlenecks for models after alignment are twofold: the compositionality gap from correct single-step results to a correct multi-step solution, and an explainable and hallucination-free intermediate single-step to multi-step reasoning path.

## 2 METHODOLOGY

**Initial Dataset**    For synthetic graph datasets, we follow Wang et al. (2023) and Zhang et al. (2024b) to choose connectivity and shortest path problems. We choose these two tasks since reasoning behind connectivity and shortest path can represent a range of real-world graph problems, including comparing if two ideas are similar or not (if those two ideas are connected) (Saha et al., 2021), solving an implicit knowledge graph question (where the model will try to reason through the shortest path between two entities) (Ding et al., 2024). On the other hand, it is easy and straightforward to verify the correctness of model responses on these two tasks with a Python program.

**Overall Pipeline**    Our approach follows a three-stage pipeline leveraging synthetic graph data for training and evaluation. We begin with an off-the-shelf model (base model in the following sections) and *optionally* perform supervised fine-tuning (SFT) on a set of synthetic graph problems. This SFT stage provides the model with exemplars of graph reasoning (including structured reasoning steps and final solutions) in a supervised manner, priming it with domain-specific patterns before reinforcement learning. After SFT (or directly from the base model if SFT is skipped), we further apply post-training alignment methods on synthetic graph tasks. Finally, we evaluate the resulting model on both held-out synthetic problems and real-world graph problems to assess reasoning generalization. This end-to-end pipeline – from base model, to optional synthetic SFT, to post-training alignment – is designed to

better understand the model's capabilities of learning and generalizing reasoning on synthetic data to real-world problems.

**Reward Design**   Our training data only contains synthetic tasks of connectivity and shortest path, whose answers can be easily verified using Python programs, so we design rule-based rewards for each task. For a more straightforward extraction of certain parts of the model's completion, we specify a special format for the response, which contains three parts: *think*, *response* and *answer*, which are enclosed by "`<think>...</think>`", "`<response>...</response>`" and "`<answer>...</answer>`" respectively. For *response* and *answer* parts, we also specify a given format of solving synthetic problems. We introduce a small format reward to encourage the model to generate clear intermediate steps resembling the format. We design two types of rewards based on this format:

- **Solution-based Reward**: We only extract the solution of the response, which is enclosed by "`<answer>...</answer>`". We then assign a reward to the response by comparing the model's solution directly with the ground truth answer. If the extraction of solution fails due to invalid format of the response, we consider it as failing the task.

- **Process-based Reward**: We extract the reasoning process and the solution to assign a proper reward to each response, which are enclosed by "`<response>...</response>`" and "`<answer>...</answer>`". For the reasoning processes enclosed by "`<response>...</response>`", we reconstruct the underlying graph using NetworkX Hagberg et al. (2008) and evaluate if all the reasoning processes are correct without hallucination or incorrect statements. For the solution, we use the same method as the Solution-based reward to evaluate its correctness.

We assign 0.2 point reward to the overall format, 0.1 point reward to process format (if applicable), and 1 point for the solution. We also introduce a penalty of -2 points that specializes in penalizing hallucinated edges or weights shown in reasoning process (and solution for shortest path). We describe their principles here while detailed implementations of reward functions, reward scores, and examples can be found in Appendix A.

## 3 EXPERIMENT SETTINGS

### 3.1 DATASETS AND EVALUATION

**Synthetic Dataset**   We directly adopt the dataset from NLGift (Zhang et al., 2024b) and build upon it. For each synthetic task, for each split of train and test sets, we take 500 questions and combine the two synthetic tasks together to construct our synthetic dataset. For the input, we choose to represent the graph using natural language by iterating all nodes first and then for each node iterating through all of the edges (and weights, if applicable). In the instruction, we also specify the desired format for output. For output, we build the ground truth response based on the reward design in Section 2. For evaluation, we use rule-based methods to extract the solution to evaluate the accuracy of each task.

**Real-World Datasets**   We use the following datasets for real-world tasks with implicit graph structures. In general, we believe there are mainly three categories related to implicit graph reasoning.

- **Multi-hop QA**: Multi-hop QA involves answering questions that need multi-step reasoning on a certain set of knowledge and information, which is strongly related to connectivity (to decide whether two concepts are related) and shortest path (to find the shortest reasoning path to connect two concepts). We adopt StrategyQA (Geva et al., 2021), Knowledge Crosswords (Ding et al., 2024), and WikiDYK (Zhang et al., 2025) for the multi-hop QA task.

- **Structured Commonsense Reasoning**: Structured commonsense reasoning incooperates commensense into questions that require structured reasoning to solve, which is also related to synthetic tasks like connectivity (to decide if two daily tasks have direct relationship) or shortest path (to decide if two ideas are supporting or countering each other). We adopt ExplaGraphs (Saha et al., 2021) and Proscript (Sakaguchi et al., 2021) for the structured commonsense reasoning task.

- **Action Planning**: Action planning requires following a set of rules and figuring out the correct steps without breaking any of the rules, which is also strongly related to connectivity (to decide

if a certain action steps can reach the final goal state) and shortest path (to generate a optimal plan which connects two states). We adopt two versions of Blocksworld (Valmeekam et al., 2023) (Planning and Verification) for the action planning task.

For all real-world datasets, we randomly sampled 1000 instances from each dataset. For most of the datasets, we follow the exact input and output format. Additional processing steps of datasets are introduced in Appendix C.

## 3.2 IMPLEMENTATION

**Models and Training**    We conduct experiments using two base LLMs to verify generality: QWEN2.5-7B-INSTRUCT (Yang et al., 2024) and LLAMA-3.1-8B-INSTRUCT (Grattafiori et al., 2024). These models are chosen to represent different model families and to ensure our findings are not specific to a single LLM. We chose two post-training methods for our experiment, one in on-policy RL and one in supervised learning. For on-policy RL, we use Group Relative Policy Optimization (GRPO) (Shao et al., 2024), an on-policy reinforcement learning algorithm that forgoes a value critic by comparing groups of model outputs to estimate advantages. For supervised supervised learning, we apply Direct Preference Optimization (DPO) (Rafailov et al., 2023), a reward-alignment technique that fine-tunes the policy on a static dataset of synthetic data generated with the base or tuned model with a high temperature of 0.9. We use a black-box LLM, Gemini 2.0 Flash (Pichai, 2024), to summarize the response if needed.

**Experiment Details**    We implement the SFT and DPO training using the HuggingFace TRL library (von Werra et al., 2020), which provides high-level APIs for transformer fine-tuning using SFT and DPO. For GRPO, we utilize the open-source VeRL toolkit (Sheng et al., 2024) to efficiently train the model with distributed rollouts and policy updates. SFT is run for 3 epochs using learning rate of 1e-5, while DPO and GRPO are run for 8 epochs on their respective training data using lr of 1e-6 and 5e-7. We use standard optimization settings and apply the AdamW optimizer in all stages (other hyperparameters are kept consistent across SFT and RL to isolate the effect of training strategy). Training and evaluation are performed on a server with 16 NVIDIA A100 GPUs (40 GB memory each), which allows us to fully fine-tune the 7B/8B models in a reasonable time frame. We use proportions z-test to test *statistical significance* for all the experiments.

## 4 RESULTS

We present main results of our paper in this section. We first evaluate on the held-out set of synthetic problems, and then select the models with good performance on synthetic tasks to evaluate on real-world datasets. While most settings of alignment methods achieved significant improvements on synthetic tasks, on real-world implicit graph reasoning tasks there are statistically significant improvements on 5 of the 8 task settings.

## 4.1 SYNTHETIC TASKS

As we adopt two reward designs and two alignment training methods with an optional SFT stage, for a single model we have eight experiment settings in total. Detailed performance results are shown in Table 1. In general, half of the settings of both models achieved significant improvements (p-val < 0.01), and 3 of the 16 settings achieved similar performance compared to SFT models. Compared with their original base models, training with SFT brings a performance increase of 156% on average, while directly training the model using alignment methods provides a performance increase of 120% on average, with GRPO performance better than average and SFT, providing a 222% performance increase. Compared with their SFT models, training additionally with alignment brings another 9% performance increase. This indicates that our alignment recipes advances synthetic graph problems compared to off-the-shelf models and existing SFT approaches.

When comparing different alignment methods, we see that GRPO achieved better performance across two models with 8 settings, with 7 out of 8 settings achieving significant improvements compared to SFT models, while DPO only achieved significant improvement in 1 out of 8 settings, which is both SFT trained first. The average performance improvement of DPO is lower than those of GRPO, with

| Reward Type | Solution-based Reward | | Process-based Reward | |
|---|---|---|---|---|
| Alignment Method | GRPO | DPO | GRPO | DPO |
| QWEN2.5-7B-INSTRUCT | | | | |
| ZERO-SHOT | 0.360 (0.602, 0.118) | | | |
| SFT | 0.700 (0.914, 0.486) | | | |
| ALIGNMENT W/O SFT | **0.950 (0.948, 0.952)** | 0.342 (0.602, 0.082) | **0.972 (0.968, 0.976)** | 0.329 (0.570, 0.088) |
| ALIGNMENT W/ SFT | **0.805 (0.962, 0.648)** | 0.698 (0.928, 0.468) | **0.967 (0.946, 0.988)** | 0.694 (0.926, 0.462) |
| LLAMA-3.1-8B-INSTRUCT | | | | |
| ZERO-SHOT | 0.232 (0.458, 0.006) | | | |
| SFT | 0.738 (0.972, 0.504) | | | |
| ALIGNMENT W/O SFT | **0.817 (0.912, 0.722)** | 0.349 (0.658, 0.040) | **0.932 (0.950, 0.914)** | 0.310 (0.616, 0.004) |
| ALIGNMENT W/ SFT | 0.592 (0.942, 0.242) | 0.785 (0.974, 0.596) | **0.935 (0.938, 0.932)** | **0.789 (0.974, 0.604)** |

Table 1: Results on synthetic tasks. The results shown are accuracies on synthetic tasks. The results are presented in the following format: overall performance (performance on connectivity, performance on shortest path). Results with significant improvement over SFT are marked with **bold**, and results with significant performance decreases are marked with grey. On-policy GRPO performs better than off-policy DPO, with better performance on synthetic tasks, better robustness to reward type and optional SFT stage.

or without the optional SFT stage. Additionally, GRPO can balance the performance of connectivity and shortest path, resulting in a smaller gap or even better performance on harder shortest path tasks.

When comparing different reward designs, while there is not a huge difference when training with DPO, results of process-based reward are constantly better than results of solution-based reward when training with GRPO, with or without the SFT stage. For instance, training with solution-based reward after SFT leads to an average performance drop of 26.7% compared to process-based reward with also high performance volatility during training, proving that deliberately designed rewards that can check the reasoning process of the model's response can lead to better results.

In general, we believe for all of the settings, except training with DPO directly on the base model, have satisfactory results on synthetic test sets, and for the following real-world evaluation, we will use these good model settings: training the base model directly with GRPO using both reward designs, and training the SFT model with both GRPO and DPO using both reward designs. We will also conduct evaluation on the base and SFT models for comparison.

## 4.2 REAL-WORLD TASKS

Real-world evaluations reveal a helpful but mixed picture: Out of the 8 dataset settings, 3 of them exhibit consistent and statistically significant improvements, and 5 where at least one alignment setting is significant. While post-training alignment on average achieve positive gains of 13.6%, it still fails to improve on certain settings or tasks. The impact of post-training alignment is task-dependent, with benefits most pronounced in problems closer to synthetic tasks such as planning and verification tasks, and some binary answer tasks, but limited improvements or even detrimental effects are observed in certain multi-hop and commonsense reasoning benchmarks. Results are shown in Table 2.

On average, comparing the zero-shot performance, directly synthetic alignment using GRPO improved real-world performance by 5.2%, while introducing synthetic SFT before synthetic alightnment using GRPO improved real-world performance by 20.3%. Compared with synthetic SFT performance, additional training using GRPO and DPO increases performance by 9.5% and 4.0% respectively.

A notable observation is that the choice of reward design (process-based vs. solution-based rewards) does not produce a consistent winner across tasks. In some evaluations the process-based reward led to better outcomes, while in others the solution-based reward (which only evaluates the final answer) proved to be more effective. On average, compared with zero-shot performance, tuning with GRPO with process-based or solution-based reward leads to 13% and 8% performance increase when skipping the synthetic SFT stage, while after synthetic SFT the performance increases are 19% and 24% respectively. This inconsistency suggests that the optimal alignment reward for synthetic graph problems may be task-dependent.

The impact of purely synthetic SFT is mixed across different datasets. Some tasks derive significant benefit from SFT alone, whereas others show little performance increase or even worse performance.

| Train Setting | StrategyQA | K-C | WikiDYK-R | WikiDYK-F | ExplaGraphs | Proscript | BW-P | BW-V | Avg. Increase |
|---|---|---|---|---|---|---|---|---|---|
| | | | | QWEN2.5-7B-INSTRUCT | | | | | |
| ZERO-SHOT | 0.702 | 0.504 | 0.067 | 0.431 | 0.829 | 0.592 | 0.114 | 0.450 | - |
| SYNTHETIC SFT | 0.615 | 0.268 | 0.069 | **0.531** | 0.668 | 0.488 | **0.226** | **0.706** | 10.6% |
| DPO-P w/ SFT | 0.622 | 0.280 | 0.070 | **0.534** | 0.689 | 0.483 | **0.202** | **0.734** | 9.7% |
| DPO-S w/ SFT | 0.624 | 0.277 | 0.067 | **0.534** | 0.690 | 0.482 | **0.192** | **0.730** | 7.9% |
| GRPO-P w/o SFT | 0.704 | 0.492 | 0.065 | **0.486** | **0.870** | 0.605 | 0.104 | 0.408 | -0.4% |
| GRPO-S w/o SFT | 0.717 | 0.523 | 0.056 | 0.374 | 0.831 | 0.521 | 0.140 | 0.482 | -0.7% |
| GRPO-P w/ SFT | 0.619 | 0.248 | 0.066 | **0.512** | 0.679 | 0.347 | **0.192** | **0.686** | 2.0% |
| GRPO-S w/ SFT | 0.612 | 0.232 | 0.070 | **0.535** | 0.743 | 0.465 | **0.182** | **0.720** | 6.2% |
| | | | | LLAMA-3.1-8B-INSTRUCT | | | | | |
| ZERO-SHOT | 0.722 | 0.508 | 0.077 | 0.297 | 0.832 | 0.554 | 0.074 | 0.562 | - |
| SYNTHETIC SFT | 0.624 | 0.296 | 0.061 | 0.195 | 0.740 | 0.460 | **0.202** | **0.660** | 6.5% |
| DPO-P w/ SFT | 0.624 | 0.326 | 0.072 | 0.229 | 0.754 | 0.451 | **0.304** | 0.556 | 25.4% |
| DPO-S w/ SFT | 0.615 | 0.321 | 0.066 | 0.227 | 0.748 | 0.457 | **0.272** | 0.558 | 18.7% |
| GRPO-P w/o SFT | 0.709 | 0.509 | 0.083 | 0.310 | 0.837 | 0.336 | **0.170** | **0.724** | 16.3% |
| GRPO-S w/o SFT | 0.695 | 0.506 | 0.081 | **0.593** | 0.848 | 0.258 | 0.052 | **0.704** | 5.6% |
| GRPO-P w/ SFT | 0.658 | 0.377 | 0.077 | 0.319 | 0.700 | 0.373 | **0.370** | 0.380 | 36.5% |
| GRPO-S w/ SFT | 0.655 | 0.313 | **0.147** | **0.360** | 0.657 | 0.404 | **0.262** | **0.690** | 36.7% |

Table 2: Performance of synthetic-only trained models on real-world tasks. Some datasets use abbreviated names: K-C stands for Knowledge Crosswords; WikiDYK-R stands for Reliability setting, while WikiDYK-F stands for factual setting; BW stands for Blocksworld, and BW-P and BW-V stands for planning and verification respectively. -P and -S are reward function choice. All results except Proscript are shown in accuracy, and Proscript result shows percentage of satisfied constraints. Significant performance increases (p-val < 0.01) compared to zero-shot performance are marked **bold**, and significant performance decreases (p-val < 0.01) are marked with grey. Avg. Increase denotes the average performance increase compared to Zero-Shot. In general, SFT and post-training alignment have mixed results on different datasets, while on five of the eight tasks there is at least one alignment setting that achieved statistically significant improvement.

For example, the Blocksworld tasks and WikiDYK factual QA saw immediate boosts from SFT with an average of 56%, which on average alignment then amplified performance by 10%. In contrast, other benchmarks did not respond as positively. In tasks like StrategyQA and Knowledge Crosswords, simply applying SFT harmed accuracy, leading to an average of 28% performance decrease comparing to zero-shot performance, which further limits the performance increase using synthetic alignment.

Results also suggest that generalization from synthetic to real-world via alignment is most successful when the target task is structurally similar to the synthetic tasks used during post-training alignment. In particular, Blocksworld planning task benefited greatly from alignment. After training on synthetic planning-like scenarios, the models were far better at solving the Blocksworld planning challenge than zero-shot performance. For instance, an alignment-tuned LLAMA-3.1-8B-INSTRUCT model achieved a dramatically higher planning success rate, increasing 129.7% compared to base or supervised-only counterparts, and an additional 83.2% performance increase when synthetic SFT-trained first.

Finally, we note there are differences between the two model families in how well they generalize with alignment. LLAMA-3.1-8B-INSTRUCT sometimes leverages post-training alignment more effectively on certain tasks; however, it also comes with a high variance of decreasing certain tasks' performance. For instance, compared to a performance increase standard error of 0.1 for QWEN2.5-7B-INSTRUCT trained with GRPO, LLAMA-3.1-8B-INSTRUCT has a standard error of more than 0.46.

Overall, both post-training aligned LLMs show mixed yet encouraging results on real-world tasks with clear successes in certain scenarios, revealing the task-specific nature of post-training alignment successes and generalization from synthetic to real-world problems.

## 5 ANALYSIS

While using purely synthetic data and designed rewards to tune the model using post-training alignment achieves good results for certain tasks, the alignment methods we use (GRPO and DPO) still cannot achieve a universal performance boost when testing on different domains. We further conduct additional analysis using QWEN2.5-7B-INSTRUCT to understand the gap between synthetic and real-world reasoning.

| QWEN2.5-7B-INSTRUCT | StrategyQA | K-C | WikiDYK-R | WikiDYK-F | ExplaGraphs | Proscript |
|---|---|---|---|---|---|---|
| ZERO-SHOT | 0.702 | 0.504 | 0.067 | 0.431 | 0.829 | 0.592 |
| SYNTHETIC SFT | 0.615 | 0.268 | 0.069 | 0.531 | 0.668 | 0.488 |
| SYNTHETIC SFT/ALIGNMENT BEST | 0.717 | 0.523 | 0.070 | 0.535 | 0.870 | 0.605 |
| REAL-WORLD SFT 6K | 0.682 | 0.785 | 0.096 | 0.676 | 0.932 | 0.851 |
| /W SYNTHETIC ALIGN | 0.671 | **0.791** | 0.092 | 0.676 | **0.936** | 0.789 |
| REAL-WORLD SFT 1.2K | 0.698 | 0.654 | 0.077 | 0.660 | 0.915 | 0.795 |
| /W SYNTHETIC ALIGN | 0.661 | 0.645 | 0.072 | 0.648 | **0.922** | 0.757 |
| REAL-WORLD SFT 0.3K | 0.708 | 0.579 | 0.069 | 0.601 | 0.901 | 0.729 |
| /W SYNTHETIC ALIGN | **0.742** | 0.449 | 0.066 | **0.641** | **0.907** | 0.638 |

Table 3: Results of mixing real-world data during the SFT stage and further tuning using GRPO on synthetic tasks. Improvements provided by synthetic RL compared to their respective REAL-WORLD SFT results are marked with **bold**. While there are 6 out of 18 settings that achieved improvements, none of the improvements is *statistically significant* (p-val < 0.01).

## 5.1 MIXING REAL-WORLD DATA

While the original goal for this work is to understand and evaluate if alignment can further generalize synthetic graph reasoning to real-world implicit graph reasoning tasks, we believe that normally we can acquire a small amount of labelled data for a real-world SFT stage with human annotation. We simulate this phase by introducing an additional SFT training using certain amount of labelled real-world data, and then we perform the alignment stage where all data instances and rewards are synthetic. While the SFT stage somehow provides the model clues about how to answer each kind of question, we are interested in finding out if further synthetic alignment training can increase the model's performance.

We derive a total of 300/1200/6000 samples with question and ground-truth pairs from the following six datasets: StrategyQA, Knowledge Crosswords, WikiDYK (Reliability), WikiDYK (Factual), ExplaGraphs, and Proscript, to simulate a small-sample SFT stage. Each real-world SFT set consists of 1/6 of the total data. We train models using SFT for 3 epochs, and then continue to use synthetic graph tasks to GRPO tune the model using process-based reward. Results are shown in Appendix B Table 3.

Out of 18 alignment results, 6 achieved improvements w.r.t. their real-world SFT setting and other baselines; however, none of the improvements is statistically significant. Also, alignment's effects are still task-dependent, without bringing a universal improvement across all real-world tasks. In general, under the current setting, even mixing a small to medium portion of real-world data, a synthetic alignment stage after that is not the golden solution to generalize reasoning beyond synthetic patterns.

## 5.2 COMPOSITIONALITY GAP

As shown in Table 2, some datasets' results after alignment training (with or without the optional SFT stage) do not show any significant performance increase. Intuitively, single-step results should generally be better, or at least not worse, than multi-step results due to the limitation of LLM's multi-step reasoning capabilities. We are interested in whether previous results indicate a limitation in knowledge, a reasoning gap between single-step and multi-step implicit graph reasoning, or any other interesting findings. We build upon three datasets and extract a group of *single-step* questions from each multi-step question to probe the synthetic-trained language model's response. Results are shown in Figure 2. Examples of single-step questions for three datasets and implementation details are presented in Appendix D.

**StrategyQA** We see out of all the correct single-step questions, there are at least 25%, or even as much as 46%, of the incorrect multi-step results across all training settings. This means that there is indeed a gap between single-step and multi-step reasoning, and even using synthetic alignment cannot solve this problem. Another interesting finding is that, out of all the correct multi-step results, at least half of the correct results come from a not-completely-correct single-step results. For instance, the model successfully answers a multi-step question 'A->C' which needs the information of single-step 'A->B' and 'B->C', but fails to answer at least one of the single-step questions. Also, the ratio of incorrect single-step out of all correct multi-step answers tends to increase with more real-world SFT data. This suggests that either the model is hallucinating the final answer, or the model is using alternative methods to solve the multi-step question. Both scenarios (using wrong/unexplainable

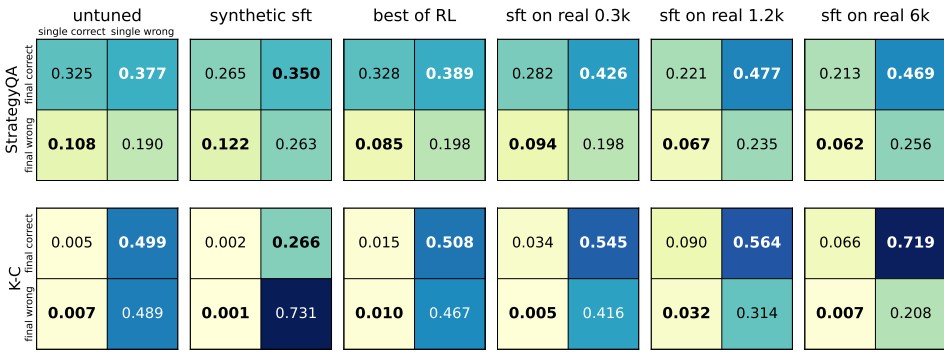

Figure 2: Proportions of cases regarding whether single steps and the final answer are correct on StrategyQA and Knowledge Crosswords. **Bolded** numbers indicate mismatched translations (i.e., wrong single steps with correct final answers and correct single steps with wrong final answers). For StrategyQA, both mismatched translations account for a significant non-zero portion of the overall result, and K-C's correct results mainly come from incorrect single-step reasoning processes.

intermediate steps to solve a multi-step question, or failing to get the correct final result even with correct intermediate steps) are not desired in this multi-step reasoning setting.

**Knowledge Crosswords** First, there is still a non-negligible portion of incorrect multi-step results out of all the correct single-step results, proving that there is indeed a reasoning gap between single-step and multi-step reasoning for Knowledge Crosswords. Second, similar to StrategyQA analysis results, there is an even larger portion of correct multi-step answers with incorrect single-step results, and with the increased size of real-world SFT, the amount of this portion increases. This further proves that, although the model reached a better overall performance for real-world dataset, it mainly comes from a scenario that the model is either hallucinating the results without correct intermediate reasoning steps, or is using an alternative reasoning path, which is not the same as human defined, to reason out the correct multi-step answer.

**Proscript** We deliberately probe the model with a single constraint per prompt for our Proscript *single-step* setting. Results are shown in Table 4. When using a significance threshold of $\alpha = 0.05$, we find that the performance difference between the two experimental groups is all statistically significant, with 5 out of 6 analysis results showing that multi-step is doing much better. On one hand, SFT using synthetic data leads to decreased multi-step reasoning capabilities on real-world tasks; on the other hand, the model's performance increases, with the help of real-world SFT or synthetic alignment, but comes with a more severe hallucination on multi-step performance, as the performance gap between single-step and multi-step is significant. We can cautiously conclude that in Proscript, performance gains rise from more severe hallucination after real-world SFT or synthetic alignment, rather than increased graph reasoning capabilities. Proscript's results align with previous results of StrategyQA and Knowledge Crosswords, and further emphasize the importance of understanding and investigating hallucination-free and explainable intermediate steps in multi-step reasoning tasks.

*In summary*, while knowledge gap may be a partial reason for the incapability of generalizing to real-world tasks, *two more significant caveats arise*. First, alignment struggles to patch the compositionality gap *from single steps to the full problem*. Second, performance increases, if any, are more often caused by *multi-step hallucination* or *misalignment with predefined multi-step reasoning steps*, as models might provide correct answers to the full problem without an accurate understanding of the underlying intermediate steps. These two limitations highlight the caveats of alignment learning with synthetic graph data and motivate solutions as future work.

## 6 RELATED WORK

**Post-training with LLMs** Large language models (LLMs) have been increasingly fine-tuned with reinforcement learning from human feedback (RLHF) to improve alignment, safety, and reasoning Ouyang et al. (2022); Bai et al. (2022); Jaech et al. (2024). Early applications of RLHF demonstrated significant gains in complex tasks such as text summarization and instruction-following (Stiennon et al., 2022; Ouyang et al., 2022) compared to previous models(Brown et al., 2020). To reduce the cost

| QWEN2.5-7B-INSTRUCT | Single-step | Multi-step | Proportion z-test p-val |
|---|---|---|---|
| ZERO-SHOT | 0.575 | 0.592 | 0.046 |
| SYNTHETIC SFT | 0.627 | 0.488 | 7.35E-61 |
| BEST ALIGNMENT: GRPO-P W/O SFT | 0.573 | 0.605 | 1.00E-4 |
| REAL-WORLD SFT 6K | 0.681 | 0.851 | 1.24E-124 |
| REAL-WORLD SFT 1.2K | 0.613 | 0.795 | 1.51E-123 |
| REAL-WORLD SFT 0.3K | 0.607 | 0.729 | 2.01E-52 |

Table 4: Proscript translation between single-step and multi-step result. Single-step means we only prompt the model to answer whether a single fact can be satisfied, while multi-step is to let the model directly generate a complete multi-step reasoning answer. For most settings, p value is small, stating that there is fundamental difference in single-step and multi-step reasoning capabilities.

of human feedback, RLAIF uses AI-generated preferences to train reward models with comparable performance (Lee et al., 2023). Most RL on LLM pipelines optimize LLM policies with on-policy algorithms, notably Proximal Policy Optimization (PPO) (Schulman et al., 2017), which originates from TRPO (Schulman et al., 2015), to iteratively maximize reward model outputs while constraining divergence from the base model. Group Relative Policy Optimization (GRPO) eliminates the need for a separate value network by normalizing rewards across batches, greatly improving training efficiency on reasoning tasks (Shao et al., 2024). On the other hand, Direct Preference Optimization (DPO), reframes preference alignment as a simple supervised objective without costly sampling (Rafailov et al., 2023). Alignment methods have substantially advanced the multi-step reasoning capabilities of LLMs (Chu et al., 2025; Guo et al., 2025a; Li et al., 2025; Kumar et al., 2024; Hu, 2025; Luo et al., 2025; Jain et al., 2025), with researchers exploring more capabilities of post-training alignment (Yue et al., 2025; Zuo et al., 2025; Shen et al., 2025; Wan et al., 2025; Wei et al., 2025; Chen et al., 2025).

**Graph Reasoning with LLMs**  Recent work has extended LLMs to reason over graph-structured data by encoding explicit graph structures, fine-tuning with graph instructions, and retrieval augmentation. A key idea is to represent graphs in an LM-readable form or code, either by linearizing graph topology into text prompts or by injecting edge lists and paths into the context (Fatemi et al., 2024; Wang et al., 2023; Han et al., 2024; Madaan et al., 2022). Beyond prompting, specialized graph instruction tuning has emerged: LLMs are fine-tuned on graph reasoning tasks with potential diverse modality to better internalize structured knowledge (Chen et al., 2024c; LUO et al., 2024; Perozzi et al., 2024; Wang et al., 2024b; Li et al., 2024b; Das et al., 2024; Tang et al., 2024; He et al., 2024b; Zhu et al., 2024; Wang et al., 2024a; Deng et al., 2024; Chen et al., 2024a), or further enhanced with graph structures (Lin et al., 2024; Chen et al., 2024b; Wu et al., 2024). Such models outperform zero-shot LLMs on tasks like knowledge graph question answering and multi-hop reasoning, demonstrating that integrating graph context can curb hallucinations and improve relational inference (He et al., 2024a; Guo et al., 2023). However, recent benchmarks suggest that while fine-tuning on synthetic graph data can teach LLMs specific patterns, these models often cannot fully transfer (Zhu et al., 2024; Chu et al., 2025; Tang et al., 2025) or struggle to transfer beyond their training distribution or generalize to real-world tasks (Zhang et al., 2024b; Guo et al., 2023). This gap has spurred new efforts to improve LLMs' graph reasoning robustness, though achieving reliable out-of-distribution generalization remains an open challenge.

## 7 CONCLUSION

In this work, we investigate using post-training alignment to generalize LLM graph learning beyond synthetic problems. We use synthetic graph tasks including connectivity and shortest path, and implement the alignment reward using rule-based rewards with two designs: process-based and solution-based reward, finding that process-based reward consistently outperforms solution-based reward. While models purely trained on synthetic problems with alignment can lead to overall synthetic performance increases and partial performance increases on real-world tasks, post-training alignment on synthetic data does not provide a universal solution to all real-world tasks. Further, we analyze the performance bottleneck of current LLMs graph reasoning, including two important caveats: failure to generalize from single-step to multi-step reasoning, and potential hallucination from single-step reasoning to multi-step reasoning. With the partial success and important findings of synthetic alignment on real-world tasks, we believe further research is in need to fully understand the compositionality and explainability of graph-related generalization of LLMs.

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

## LIMITATIONS

**Adopted Methods** We only adopt two representative post-training alignment methods (GRPO and DPO) on LLMs, while there are still a wide range of on-policy optimization methods (Zhang et al., 2024a; Schulman et al., 2017; Yuan et al., 2025b; Yu et al., 2025; Guo et al., 2024; Rosset et al., 2024; Agarwal et al., 2024; Ahmadian et al., 2024) and off-policy optimization methods (Xu et al., 2024; Ethayarajh et al., 2024; Meng et al., 2024) with possible model-based rewards (Uesato et al., 2022; Khalifa et al., 2025). With that in mind, in general, we do see that different methods are enhancing training efficiency and stability and should not impact the overall results of our work greatly.

We also notice there are different methods besides adopted post-training alignment, aiming to improve general reasoning capabilities of LLMs, including test-time scaling (Muennighoff et al., 2025; Snell et al., 2024; Setlur et al., 2025; Li et al., 2025; Zuo et al., 2025; Huang et al., 2025; Yuan et al., 2025a) and sub-response level reward, supervision or verification (Xiong et al., 2024; Uesato et al., 2022; Lightman et al., 2023). We leave experimenting using these methods for future work.

**Adopted Models** We only adopt two representative open-source models (QWEN2.5 and LLAMA3.1 with relative small size (7B and 8B respectively) (Li et al., 2024a). While there are still a wide range of open-source LLMs (Grattafiori et al., 2024; Yang et al., 2024; Jiang et al., 2023; Liu et al., 2024a; Team et al., 2024) and close-source LLMs (ant; Guo et al., 2025b; dee; ope; Jaech et al., 2024; Hurst et al., 2024; Pichai, 2024) with different model sizes, we do believe the ultimate goal is to achieve a universal and general good performance across all models and all sizes, especially with potential of fully utilizing synthetic data on real-world tasks, and reliable and huallicination-free reasoning steps in multi-step reasoning tasks.

**Real-World Datasets** We only adopt 8 real-world tasks within 3 main categories (multi-hop QA, commonsense reasoning, and action planning). There are still plenty of datasets that belong to these three categories (Mavi et al., 2024; Davis, 2023; Talmor et al., 2021; Ghazal et al., 2013; Yang et al., 2025; Liu et al., 2024b; Choi et al., 2024) and other reasoning related datasets and domains (Hendrycks et al., 2021; Cobbe et al., 2021; Tong et al., 2024; Qiu et al., 2025; Jimenez et al., 2023; Zhuo et al., 2024; Sui et al., 2024; Xiong et al., 2023; Sprague et al., 2023; Xin et al., 2024; Wen et al., 2024). We also notice that our selected datasets may not thoroughly represent real-world datasets and tasks perfectly, but in general they still have a large gap with purely synthetic data generated by algorithms and symbolic representations.

## REPRODUCIBILITY STATEMENT

We present hyperparameter details, training settings, and other experiment details in Section 3 and Appendix A to C. We provide experiment data and code anonymously at `https://anonymous.4open.science/r/Graph_RL-BF08/readme.md` and will make it publicly available upon acceptance.

## A REWARD IMPLEMENTATION DETAILS

**Reward Function** We provide a high-level algorithm of connectivity reward and shortest path reward function in Algorithm 1 and Algorithm 2. The actual implementation follows the overall logic given by the algorithm but may has minor differences. We further define the following reward values for evaluating the agent's response:

- **Correct answer reward:**
  $r_{correct\_answer} = +1$
- **Incorrect answer penalty:**
  $r_{incorrect\_answer\_penalty} = 0$
- **Hallucination penalty:**
  $r_{hallucination\_penalty} = -2$
- **Correct reasoning step format reward:**
  $r_{correct\_step} = +0.1$

- **Incorrect reasoning penalty:**
  $r_{\text{incorrect\_step\_penalty}} = 0$

- **Correct format reward:**
  $r_{\text{format\_reward}} = +0.2$

- **Format error penalty:**
  $r_{\text{format\_penalty}} = 0$

---

**Algorithm 1** Connectivity Reward

---

**Require:** $R$: agent's full response with step-by-step reasoning and a final connectivity answer.
**Require:** $G$: graph definition (nodes and edges of the given graph).
**Require:** $A$ and $B$: two nodes in the given graph.
**Require:** *type*: either "process" or "solution".
**Require:** *ground_truth*: ground truth connectivity ("yes" if $A$ and $B$ is connected, "no" otherwise).
**Ensure:** $r$: total reward for the response.
   $r \leftarrow 0$                                                            ▷ initialize reward score
   *// Format checking*
   **if** response $R$ is not in the expected format (*e.g.*, missing reasoning steps or no clear final answer)
   **then**
      $r \leftarrow r + r_{\text{format\_penalty}}$                                ▷ penalize formatting issue
   **else**
      $r \leftarrow r + r_{\text{format\_reward}}$                                ▷ reward correct format
   **end if**
   $(\textit{thought}, \textit{reasoningProcess}, \textit{answer}) \leftarrow \text{SEPARATERESPONSE}(R)$
   *// Evaluate each reasoning step for correctness and hallucinations*
   **if** $\textit{type} = $ "process" **then**
      $\textit{rewards} \leftarrow []$                                      ▷ initialize an empty list
      **for each** step $s$ **in** *reasoningProcess* **do**
         **if** $s$ references any node or edge not present in $G$ **then**
            Append $r_{\text{hallucination\_penalty}}$ to *rewards*        ▷ penalize hallucinated graph elements
         **else if** $s$ is a correct logical statement about $G$ **then**
            Append $r_{\text{correct\_step}}$ to *rewards*                 ▷ reward a correct reasoning step
         **else**
            Append $r_{\text{incorrect\_step\_penalty}}$ to *rewards*       ▷ penalize a step with incorrect format
         **end if**
      **end for**
      $r_{\text{avg}} \leftarrow \text{Average}(\textit{rewards})$                         ▷ average the process reward
      $r \leftarrow r + r_{\text{avg}}$
   **end if**
   *// Final answer correctness*
   **if** *answer* matches *ground_truth* (*e.g.*, correctly says "yes" or "no") **then**
      $r \leftarrow r + r_{\text{correct\_answer}}$                               ▷ reward correct conclusion
   **else**
      $r \leftarrow r + r_{\text{incorrect\_answer\_penalty}}$                   ▷ penalize incorrect conclusion
   **end if**
   **return** $r$

---

## B   ADDITIONAL TRAINING IMPLEMENTATION DETAILS

**LLAMA3.1-8B-INSTRUCT Training Details**   Experiments on LLAMA3.1-8B-INSTRUCT experience high level of volatility, shown in Table 1. While we try to tune hyper-parameters to get best results using LLAMA3.1-8B-INSTRUCT, some settings (especially with GRPO) trained to collapse, leading to worse results compared to zero-shot on synthetic tasks. For three of the previous settings in Table 1 and Table 2 using LLAMA3.1-8B-INSTRUCT, including GRPO-P W/O SFT, GRPO-S W/O SFT, GRPO-S W/ SFT, we train around 3.2 epochs (rather than standard 8 epochs) using saved checkpoints for better representation of model's performance.

---

**Algorithm 2** Shortest Path Reward

---

**Require:** $R$: agent's full response with step-by-step reasoning and a final shortest path answer.
**Require:** $G$: weighted graph (nodes, edges, and edge weights).
**Require:** $s, t$: start and end nodes for shortest path query.
**Require:** *type*: either "process" or "solution".
**Require:** *ground_truth*: shortest path length between $s$ and $t$ in $G$.
**Ensure:** $r$: total reward for the response.
  $r \leftarrow 0$
  *// Format checking*
  **if** response $R$ is not in the expected format (*e.g.*, missing reasoning steps or no clear final answer) **then**
    $r \leftarrow r + r_{\text{format\_penalty}}$
  **else**
    $r \leftarrow r + r_{\text{format\_reward}}$
  **end if**
  $(thought, reasoningProcess, answer) \leftarrow \textsc{SeparateResponse}(R)$
  **if** *type* = "process" **then**
    $rewards \leftarrow [\,]$
    **for each** step $s$ **in** *reasoningProcess* **do**
      **if** $s$ references any node or edge not in $G$ **then**
        Append $r_{\text{hallucination\_penalty}}$ to *rewards*
      **else if** $s$ describes a valid fact or update consistent with shortest path logic **then**
        Append $r_{\text{correct\_step}}$ to *rewards*
      **else**
        Append $r_{\text{incorrect\_step\_penalty}}$ to *rewards*
      **end if**
    **end for**
    $r_{\text{avg}} \leftarrow \text{Average}(rewards)$
    $r \leftarrow r + r_{\text{avg}}$
  **end if**
  *// Final answer evaluation*
  $(path, length) \leftarrow \textsc{ParseShortestPath}(answer)$
  **if** *path* or *length* is missing or ill-formed **then**
    $r \leftarrow r + r_{\text{format\_penalty}} + r_{\text{incorrect\_answer\_penalty}}$
  **else if** *path* contains non-existent nodes or edges in $G$ **then**
    $r \leftarrow r + r_{\text{hallucination\_penalty}}$
  **else if** *length* = *ground_truth* **and** *path* is valid in $G$ **then**
    $r \leftarrow r + r_{\text{correct\_answer}}$
  **else**
    $r \leftarrow r + r_{\text{incorrect\_answer\_penalty}}$
  **end if**
  **return** $r$

---

| Task | Example Prompt | Summarization Prompt |
|------|----------------|----------------------|
| StrategyQA | Please think step by step and then answer the following question with either yes or no: Could you make the kitchen 'holy trinity' without celery? | The following paragraph is the answer to the question. Summarize the paragraph's answer using either "YES", "NO" or "UNKNOWN". Question: {prompt} Paragraph: {response} |
| K-C | Please select one option that satisfies all the constraints in the question. Please note that the 3 words in each option are from blank 1 to 3. The question is: blank 2 actedIn Casper_(film), blank 2 actedIn The_Man_Who_Cried, Rose_McGowan actedIn blank 3, blank 1 actedIn blank 3, blank 1 actedIn The_Man_Who_Cried. Options: A. Brandon_Routh, Christina_Ricci, Robinson_Crusoe_(1997_film) B. Marjorie_Rambeau, Sin\u00e9ad_Cusack, Robinson_Crusoe_(1997_film) C. John_Turturro, Christina_Ricci, Monkeybone D. Marjorie_Rambeau, Christina_Ricci, Robinson_Crusoe_(1997_film) Please think step by step. Your last sentence should give a single letter from A to D. | The following paragraph is the answer to a the question. Decide if the paragraph gives a definite answer of yes or no, and what the answer is. Summarize the paragraph's answer using either "A", "B", "C", "D" or "UNKNOWN". Put your answer in double asterisks, like **A/B/C/D/UNKNOWN**. Question: {prompt} Paragraph: {response} |
| WikiDYK-R | What type of Japanese girls' comics were mostly written by men until the 1970s? | Does the following paragraph mention the following word (doesn't have to be exact match)? Answer using either "YES", "NO" or "UNKNOWN". Paragraph: {response} Word: {ground_truth} |
| WikiDYK-F | until the 1970s, most Shojo manga (Japanese girls' comics ) were written by men. Is this statement true or false? | N/A |
| ExplaGraphs | Please judge if the following two sentences support each other or counter each other: "Bad foster care parents has negative effect on a kid" and "When parent of foster homes are not good it tends to have a traumatizing effect on a child". Please think step by step, and then respond either "support" or "counter": | N/A |
| Proscript | Identify the logical order of all the following steps to achieve the following goal. Note that the numbering of the steps does not indicate their execution order, and your response should include all steps. Please think step by step. Goal: Get glass of milk Steps: step0: close the fridge; step1: retrieve the milk from the fridge; step2: retrieve a glass from the cabinet; step3: walk toward the kitchen; step4: open the door to the fridge; step5: pour the milk into the glass; step6: put the milk back in the fridge; step7: decide to Get glass of milk; step8: Get glass of milk Format your response as a sequence, using "->" to separate (e.g., "step8->step4->step3"). | summarize the response so that it follows the following format (arrow linked with no space in between): stepA->stepB->stepC->...->stepK, where A, B, C, K are single digit numbers. Do not include or use any other texts. If the response doesn't include any steps, respond with "UNKNOWN". The response is: {response} |
| BW-P | I am playing with a set of blocks where I need to arrange the blocks into stacks. Here are the actions I can do (all rules) Your task is to generate a plan for the following goal. The plan should be valid and can be executed in the blocksworld environment. The plan should be a sequence of actions that can be performed to achieve the goal. As initial conditions I have that, the red block is clear, the yellow block is clear, the hand is empty, the red block is on top of the blue block, the yellow block is on top of the orange block, the blue block is on the table and the orange block is on the table. My goal is to have that the orange block is on top of the red block. My plan is as follows: [PLAN] | The following paragraph is a plan to solve the blocksworld problem. Verify if the plan is executable and can reach the goal while following all the rules. Summarize the paragraph's answer using either "VALID" or "INVALID". Problem: {prompt} Paragraph: {response} |
| BW-V | I am playing with a set of blocks where I need to arrange the blocks into stacks. Here are the actions I can do (all rules) Your task is to verify if the following plan is valid or invalid, and give out an explanation of why it is valid or invalid. The plan is valid if it can be executed in the blocksworld environment while following all rules and the outcome is the same as the goal. The plan is invalid if it cannot be executed or the outcome is different from the goal. As initial conditions I have that, the red block is clear, the blue block is clear, the yellow block is clear, the hand is empty, the blue block is on top of the orange block, the red block is on the table, the orange block is on the table and the yellow block is on the table. My goal is to have that the orange block is on top of the blue block. My plan is as follows: [PLAN] unstack the blue block from on top of the orange block stack the orange block on top of the blue block pick up the orange block [PLAN END] [VERIFICATION] | The following paragraph is the answer to a certain question. Summarize the paragraph's answer using either "VALID", "INVALID" or "UNKNOWN". Paragraph: {response} |

Table 5: A full list of example prompts and example summarization prompt (if applicable) of real-world tasks. All example prompt are exactly the same with the actuall prompt for the LLM, except BW-P and BW-V rules part is omitted for better display. Most of the datasets follow their original implementation, while we change K-C, BW-P and BW-V a little for aligned evaluation.

| Task | Multi-step Q&A | Single-step Q&A | Analysis |
|------|----------------|-----------------|----------|
| StrategyQA | *User:* Does Adam Sandler skip celebrating Easter? *Assistant:* ...Based on my knowledge, I couldn't find any specific information about Adam Sandler skipping Easter celebrations. ✗ | *User:* Easter is usually celebrated by people of which religion? *Assistant:* Easter is primarily celebrated by Christians, specifically those who follow the Christian faith. ✔ *User:* What is Adam Sandler's religion? *Assistant:* Adam Sandler is Jewish.✔ | incorrect final multi-step answer, but intermediate steps are correct, possibly single-to-multi-step reasoning gap |
| K-C | *User:* ...The question is: blank 2 actedIn Casper_(film) blank 2 actedIn The_Man_Who_Cried ... Options (blank 1, blank 2, blank 3): ... C. John_Turturro, Christina_Ricci, Monkeybone ... *Assistant:* ...The answer is C. ✔ | *User:* Please decide if the following information is true or false: Christina_Ricci actedIn Casper_(film). *Assistant:* true ✔ *User:* Please decide if the following information is true or false: Christina_Ricci actedIn The_Man_Who_Cried. *Assistant:* false ✗ (3 remaining, all correct) | correct multi-step final answer but incorrect intermediate single-step answers, hallucination or unexplainable intermediate steps |
| Proscript | *User:* Identify the logical order of all the following steps to achieve the following goal. Note that the numbering of the steps does not indicate their execution order, and your response should include all steps. Please think step by step. Goal: Get glass of milk Steps: step0: close the fridge... *Assistant:* To achieve the goal of getting a glass of milk, the logical order of steps is: step7->step3->step4->step0->step1->step2->step5->step6 (*multi-step achieved 6 out of 9 constraints*) | *User:* We have a target of 'Get glass of milk'. We already completed certain steps. Now there are two intermediate next steps we need to take: step 1: retrieve the milk from the fridge step 2: open the door to the fridge. We plan to do step 1 before step 2. Is this the right order to execute these two steps in order to achieve the target? You should answer yes or no. *Assistant:* Yes. ✗ (8 remaining, 5 of them are correct, and 3 of them are incorrect, *single-step achieved 5 out of 9 constraints*) | single-step reasoning is worse than multi-step reasoning, possibly hallucination or alternative single-step reasoning steps |

Table 6: Examples of multi-step and single-step prompts and model's responses. Some prompts are shortened for readability. Results shown in this table and previous analysis show that two shortcomings of model's real-world reasoning capabilities: failure to generalize from single-step to multi-step, and failure to provide an explainable and correct reasoning process due to incorrect single-step response.

## C    REAL-WORLD DATASET IMPLEMENTATION DETAILS

**Dataset Processing**    For all real-world datasets, we first randomly sample 1,000 samples as test set, and then leave the remaining for potential training set used for analysis. Several different implementations include:

- **Knowledge Crosswords**: We transform this dataset from a multiple blank choice question (for instance, a question will have a separate set of choices for each blank, making it hard to extract the answer) to a common multiple-choice question with 4 choices.

- **Blocksworld**: We remove the one-shot prompt style to match with other tasks settings. Also, Blocksworld dataset has only 500 instances per setting. Thus, during analysis of mixing real-world data, no Blocksworld data is used to train the model.

We provide examples of prompts, summarization prompts (if applicable) for each real-world dataset in Table 5.

## D    ADDITIONAL ANALYSIS

In this section, we provide more analysis on experiment results, with additional analysis on Proscript, implementation details of compositionality gap, and synthetic SFT's role in real-world performance.

### D.1    COMPOSITIONALITY GAP IMPLEMENTATION DETAILS

**StrategyQA**    StrategyQA involves using at least two steps to solve a single question. The dataset provides intermediate questions for us to decompose the multi-step question. For example, to answer a multi-hop question like "Does Adam Sandler skip celebrating Easter?", the dataset provides a series of single-step questions, including: 1. "Easter is usually celebrated by people of which religion?", 2. "What is Adam Sandler's religion?", 3. "Is #1 different from #2?", and related facts, including ["Adam Sandler is Jewish." and "Jewish religious people do not celebrate Easter."]. We prompt the LM with single-step questions with all except any with the "#", and use a LM as judge to decide if the model's response is included in the list of facts. We then map all the single-step responses to the original multi-step question, and consider if there is any compositionality gap or hallucination of final answer. An example of single-step and multi-step prompt is shown in Table 6.

**Knowledge Crosswords**    For Knowledge Crosswords, we insert all the correct answers to all blanks and prompt the model if each constraint is correct or not, to build a *single-step* setting for this dataset. We then map the single-step results back to their original question to investigate if there is a similar pattern with StrategyQA. For instance, for a multi-step question with five constraints and three blanks, we insert the correct answer to all blanks and prompt the language model with *single-step* questions of all the constraints.

**Proscript**    Proscript is a task where a goal of a daily task is proposed and the model is prompted to arrange several steps into the right order. For instance, let the goal be "Get a glass of milk", and several intermediate steps are "step1:retrieve the milk from the fridge", "step2:pour the milk into the glass" and "step3:retrieve a glass from the cabinet". A possible right sequence of actions is "step3->step1->step2", since a person needs a glass (step3) and the milk (step1) ready to pour the milk into the glass (step2), which we denote as two constraints for this single question (step3 before step2, step1 before step2). For the multi-step setup, we prompt the model to generate the complete action sequence, which in expectation should contain all steps mentioned in the intermediate steps. For instance, if the model's response is "step1->step3->step2", it satisfies both constraints, while "step1->step2->step3" only satisfies one constraint of step1 before step2. Results for multi-step setup are calculated as the number of all satisfied constraints from all samples divided by the number of all the constraints, and shown in Table 2. For the single-step setup, we are only prompting the model with a single constraint. For instance, we prompt the model "The goal is to get a glass of milk. Should step3:retrieve a glass from the cabinet be executed before step2:pour the milk into the glass?" and the correct answer should be "yes". An example of single-step and multi-step is shown in Table 6.

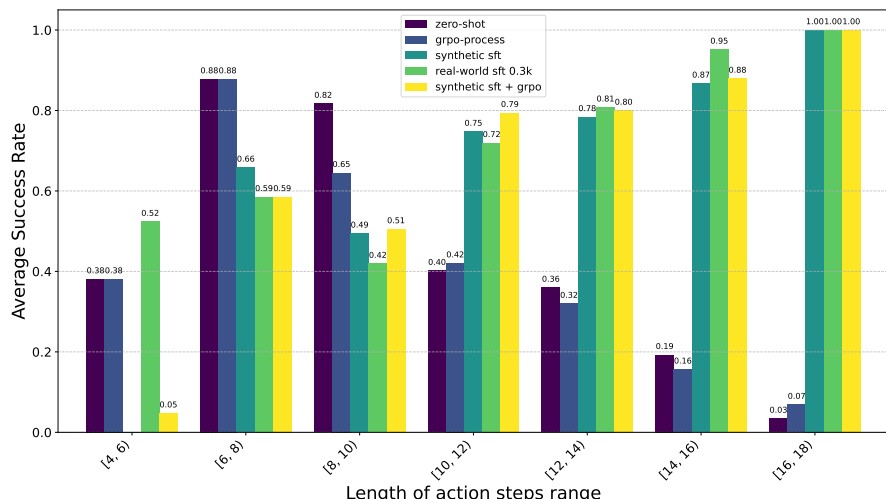

Figure 3: Performances on Blocksworld Verification task grouped by the length of action steps needs to be verified. With synthetic SFT and real-world SFT, the performance on verifying plans with longer steps gradually increases, while using alignment alone cannot produce the result.

## D.2 SYNTHETIC SFT'S ROLE ON SYNTHETIC TASKS

SFT stage acts as different roles for different alignment methods for synthetic tasks, as shown in Table 1. For on-policy method GRPO, it may decrease the robustness of the trained model as 2 out 4 settings (SFT + GRPO) cannot achieve comparable results compared to those GRPO trained models trained without SFT stage, leading to limitation of the model's performance if SFT stage is enforced before GRPO. However, for off-policy method DPO, SFT can somehow bring advantage to the model, with 2 out of 4 settings (SFT + DPO) achieved better results compared to only those only trained using SFT, while those models directly trained using DPO cannot achieve better results than SFT models.

## D.3 SYNTHETIC SFT'S ROLE ON REAL-WORLD TASKS

While the synthetic SFT stage has mixed results for different datasets, performance on Blocksworld has somewhat satisfactory results, and models trained with synthetic SFT achieved most of the good results. We dive in and analyze the verification results generated by models grouped by the length of action steps needed to achieve the target goal.

Results in Figure 3 show surprising results: with the help of synthetic SFT and real-world SFT, the model's performance on long sequence plans increases, while only using alignment methods cannot provide this result. We believe that this result comes from the help of strong supervision using strict format SFT data without any hallucination errors. This result also strengthens the idea that providing correct reasoning steps for the correct tasks should generally be helpful, while the best method for supervision still require future research.

## D.4 CORRELATION ANALYSIS

We investigate the correlation between various performance and reward metrics. First, to better understand the reward design's effect, we analyze the correlation between the synthetic task's accuracy performance and the reward score achieved by the model. The Pearson correlation coefficient is 0.951: our reward design is highly correlated with the performance of the model, proving the reward function's effectiveness. Second, we analyze the correlation between synthetic performance and real-world performance (denoted by the mean performance of all real-world tasks). The correlation is -0.336, which aligns with our finding that performance gains provided by synthetic training are mixed across models and different tasks. Detailed results are shown in Figure 4.

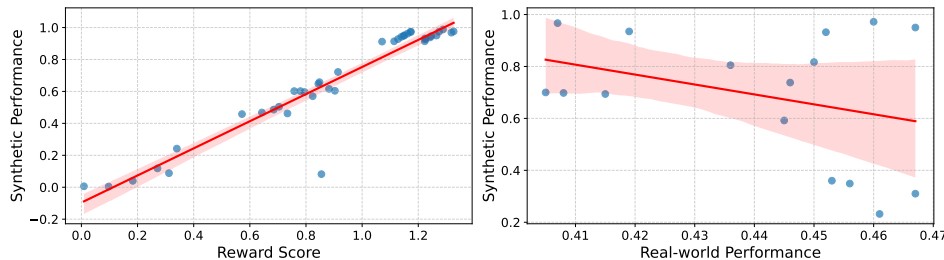

Figure 4: (left) Correlation between the synthetic tasks' performance and the reward score. The high correlation demonstrates the effectiveness of our reward design. (right) Correlation between performance on synthetic tasks and real-world tasks, showing mixed performance gains provided by synthetic training across different training methods. The red shaded area indicates the 95% confidence interval for the regression estimate.

