# OpenReview forum: "Generalizable LLM Learning of Graph Synthetic Data with Post-training Alignment"
_ICLR.cc/2026/Conference — ICLR 2026 Conference Withdrawn Submission_

### Official Review · Reviewer_bAPa · 2025-10-15

**Soundness:** 2
**Presentation:** 2
**Contribution:** 2
**Rating:** 2
**Confidence:** 5

**Summary:**

This work trains LLMs with GRPO/DPO on synthetic graph connectivity and shortest-path problems, and finds that the reasoning ability can generalize to real-world multi-hop QA and structured planning. The further analysis shows the misalignment between single-hop and multi-hop results.

**Strengths:**

The paper shows the generalization of graph-connectivity and shortest-path problems to real-world multi-hop reasoning tasks. It is an interesting direction, and the analysis of their relationships is worth exploring.

**Weaknesses:**

1. "Unlocking generalizable graph reasoning by post-training alignment on synthetic data" is not a new idea, as already verified by previous papers [1,2].

2. The novelty of training methods (solution-based or process-based GRPO/DPO) is lacking.

3. The analysis of the relationship between synthetic graph reasoning and real-world multi-hop reasoning is lacking. The analysis in Section 5 can not explain the experimental results in Section 4. (See details in Question 3)



[1]. G1: Teaching LLMs to Reason on Graphs with Reinforcement Learning. Guo Xiaojun, et al.https://arxiv.org/abs/2505.18499

[2]. Graph-R1: Unleashing LLM Reasoning with NP-Hard Graph Problems. Wang Yuyao, et al.https://arxiv.org/pdf/2508.20373

**Questions:**

1. What's the detail of process-based rewards in GRPO/DPO? In other words, how do you "reconstruct the underlying graph using NetworkX and evaluate if all the reasoning processes are correct without hallucination or incorrect statements"? If you use a stronger model Gemini 2.0 Flash, as indicated in line 180, it is unfair to compare it with other baselines that do not rely on external models.

2. How do you evaluate the results in Table 1 and Table 2, e.g., LLM-as-Judge or regex match? My concern is that the base model might be under-evaluated for the inability to strictly follow the "\<answer>...\</answer>" format when extracting answers.

3. Although the multi-hop QA can be understood as graph reasoning in **humans'** subjective perception, it's unknown whether they are also modeled as underlying graphs in LLMs. Previous works explore whether LLMs latently perform multi-hop reasoning, but the findings are contextual. Moreover, the results in Section 6 of this paper also strongly verify the misalignment between single-step results and the final results in multi-hop QA before and after training. Therefore, how (and even whether) training on tasks such as graph connectivity can help multi-hop reasoning is unclear.

[3]. Do Large Language Models Latently Perform Multi-Hop Reasoning? Sohee Yang, et al. https://arxiv.org/abs/2402.16837

[4]. Do Large Language Models Perform Latent Multi-Hop Reasoning without Exploiting Shortcuts? Sohee Yang, et al.https://arxiv.org/pdf/2411.16679



Overall, I suggest that the authors could transfer the focus of the paper from **showing** generalizable graph reasoning by post-training on synthetic data (already verified by previous works) to **analyzing** when and how synthetic graph reasoning can help real-world multi-hop/structured reasoning, which is under-explored and valuable in this domain.

---

### Official Review · Reviewer_NPgR · 2025-10-17

**Soundness:** 2
**Presentation:** 2
**Contribution:** 2
**Rating:** 2
**Confidence:** 4

**Summary:**

This paper investigates whether post-training alignment methods can help LLMs generalize from synthetic graph tasks (connectivity and shortest path) to real-world tasks with implicit graph structures. The authors design two reward types and apply alignment to both base models and models already fine-tuned on synthetic data. Experiments with Qwen2.5-7B and LLaMA-3.1-8B on synthetic tasks and eight real-world datasets show that post-training alignment improves synthetic task accuracy substantially.

**Strengths:**

1. I agree with the claim that using LLM to address tasks like the shortest path is not meaningful. I'm glad that this paper addresses the issue.
2. The idea to use RL to extend LLM's ability from small tasks to real-world applications are amazing.

**Weaknesses:**

1. **Missing critical baselines and graph-specific LLM comparisons**

The paper only compares against zero-shot and SFT baselines from the same model family. It omits comparisons with:
- Graph-specialized LLMs cited in related work (G-Retriever, GraphWiz, and GraphRAG family for Multi-hop QA).
- Strong prompting baselines (chain-of-thought with synthetic examples, PiVe's iterative verification)
- Multi-task training jointly on synthetic+real-world data
- NLGift with extended training as an SFT upper bound \
It is necessary to determine whether the idea of the transit model learned from graph reasoning actually performs better in real-world tasks.

2. **Severe inconsistency across models and datasets undermines reliability**
- **Model-level**: Table 1 shows GRPO+SFT helps LLaMA  but hurts Qwen. DPO shows opposite patterns between models.
- **Task-level**: Table 2 shows StrategyQA degrades (0.702→0.612-0.624), Knowledge Crosswords drops 28% after SFT, while Blocksworld-P improves 129%. No explanation for why identical training produces opposite effects.

These inconsistencies suggest the method lacks robustness and may not learn generalizable reasoning. Only 5/8 tasks show any significant improvement.

3. **Limited technical novelty**
**(Minor)** The approach applies existing standard SFT+RL methods pipelines without graph-specific innovations.
While understandable, the novelty still degrades a little.

4. **Insufficient analysis**
Details are missing:
- **No learning curves**.
- **Extremely small data**: Only 500 instances/task (1000 total) for training. No data scaling experiments to test if performance is data-limited.
- **Limited task diversity**: No ablation testing whether more tasks (cycle detection, topological sort) improve generalization. It is concerning because the current improvement is really unstable.

5. **Presentation**
Conclusions are mixed throughout the paper. While the dataset-specific analysis in Section 5 is detailed, it would be better to see summary words or informative comparisons that help capture the key discoveries.

**Questions:**

1. *Intuitively, single-step results should generally be better, or at least not worse, than multi-step results due to the limitation of LLM’s multi-step reasoning capabilities.*

Why does the above claim hold?

2. How to understand the −0.336 correlation between synthetic and real-world performance in Figure 4? What is the experiment setup?
Is it better to compare the LLM in both tasks to see the correlation?
If negative correlated, why RL on synthetic tasks help?

---

### Official Review · Reviewer_2sVL · 2025-10-23

**Soundness:** 2
**Presentation:** 2
**Contribution:** 2
**Rating:** 2
**Confidence:** 4

**Summary:**

This paper presents a thoughtful and well-motivated exploration of post-training alignment to enhance the generalizability of large language models (LLMs) trained on synthetic graph data. The proposed framework using GRPO and process-based rewards introduces a novel way to connect structured reasoning from synthetic graph tasks to open-ended real-world problems such as multi-hop QA and planning.

**Strengths:**

The experiments are extensive, covering both synthetic and natural datasets, and the analyses demonstrate careful empirical rigor. The use of process-based rewards is commendable, as it reflects a shift from mere output correctness to reasoning faithfulness, a crucial step for trustworthy LLM reasoning.

**Weaknesses:**

1. The evaluated graphs are small. Scalability to larger, common real-world networks is not demonstrated.

2. Not all the tasks that could be modeled graph should be modeled as graph. These tasks can be addressed by LLMs and why do we need to model them as graph?

3. Although the paper claims to study “graph reasoning,” it does not address any explicit graph problem in the conventional sense (e.g., shortest-path search, graph coloring, or subgraph matching). All evaluations are performed on language-based tasks with implicit graph structures, rather than on benchmarked graph datasets.

**Questions:**

1. How does the proposed method scale to larger or real-world graphs (e.g., thousands of nodes)?

2. What are the main contributions else than the post-training? The contribution looks limited as it has the design of two reward functions only.

---

### Official Review · Reviewer_qWNn · 2025-10-26

**Soundness:** 3
**Presentation:** 3
**Contribution:** 3
**Rating:** 6
**Confidence:** 3

**Summary:**

This paper explores post-training of LLMs on synthetic graph tasks (connectivity and shortest paths). The ultimate goal is to test whether such pre-training enhances capabilities in other reasoning problems that may need similar "path-finding" skills. As example tasks, the paper uses Multi-hop QA, Structured Commonsense Reasoning, Action Planning. They explore 2 reward methods for RL (with optional SFT): solution-based, i.e., a reward for the correct solution, and process-based, i.e., rewards for "correct" intermediate reasoning steps. For each, they try GRPO and DPO, and two LLMs (Qwen2.5-7B-Instruct, Llama-3.1-8B-Instruct).

The findings are that the post-training helps improve capabilities for the synthetic tasks, especially with the process-based reward and GRPO.

For real-world tasks, the picture is somewhat more mixed and task-dependent. Interestingly, the reward model does not seem to play a major role in this transfer setting. The authors also find that in several cases, the model returns a correct solution but cannot do 1-step parts towards such solutions, i.e., it may take a different reasoning route or shortcut.

**Strengths:**

- I think the research question is interesting, of how strong the effect of synthetic post-training tasks can be for more real-world reasoning tasks. This is an important question in general, and for graphs in particular. Importantly, the synthetic tasks are fairly easy to generate and verify.

- The findings on the relative insignificance of the process-based reward, despite its benefit for synthetic tasks, is interesting. Moreover, the findings on compositionality, i.e., that in many cases the model generates correct multi-step solutions but cannot properly do a single step that we would associate with a correct multi-step reasoning process/solution. This gives food for thought for follow-up works.

- The paper tests two reward strategies which both make sense and are relevant. In particular the process-based rewards are interesting.

**Weaknesses:**

- The writing of the main text could be clearer in certain parts. E.g., the two reward strategies should be explained at least by summary in words in the main paper, including that each correct step gets a reward. That was clear to me only after looking at the appendix. Please also see my questions below.
As another example, what the relation between the 1-step and multi-step tasks is was only clear after looking at the appendix.

- The novelty of the paper is not entirely clear. The evaluation of post-training with GRPO and the solution-based reward has been done in other works, too (e.g., [1]). By looking at the related works section, especially the second half of the paragraph on "Graph Reasoning with LLMs", it is not clear what exactly in this paper is different from prior work, i.e., what exactly are the new insights. E.g., compared to (Zhu et al., 2024; Chu et al., 2025; Tang et al., 2025, Zhang et al., 2024b; Guo et al., 2023).

- For some procedures, the reason for exactly this choice of procedure was not so clear to me -- see my question below on Section 5.1 and the rewards.

- What would be the effect of other graph tasks? This paper looks at only 2 tasks jointly. It does evaluate them thoroughly, but it would be interesting to see whether a wider range of synthetic tasks may change the findings.

[1] Guo et al., 2025. G1: Teaching LLMs to Reason on Graphs with Reinforcement Learning.

**Questions:**

1. Why is there a different process-based reward for connectivity and shortest path? Couldn't one just do one that checks the validity of the next step, regardless of the task?

2. When using process-based rewards, how can one avoid that the model artificially inserts irrelevant but correct reasoning steps in the reasoning chain (reward hacking)? The condition "s is a correct logical statement about G" does not necessarily prevent that. The model could also just repeat the same correct step over and over. the Shortest Path reward may take care of this, but it may be a problem for connectivity? Is it?

3. How big are the graphs you are using for training and testing? (It would be good to state it in the paper's main text.)

4. Somewhat related to the compositionality: if you train with the solution-based reward, what is the CoT like at test time (for synthetic and real-world task)? Would it still run a mostly valid algorithm or hallucinate? I mean here what it actually does in the multi-step solutions.

5. Section 5.1: what was the reason for training with real-world examples first, and then synthetic? What if you would jointly train on real-world and synthetic (interleaved), or do synthetic first? As a conjecture, the real-world data may help more after the synthetic training emphasized relevant "circuits" or mechanisms for path-based reasoning.


minor comments (no need to reply to these):

- Figure: the illustration of the process based reward can be misleading -- the wrong CoT could be interpreted as doing a Bellman Ford or Dijkstra style approach, so it is not wrong per se. It just needs more steps.
- line 156: "incooperates": maybe transforms?
- line 177/178: "supervised supervised"
- line 962: may has -> may have

---

### Official Review · Reviewer_wCD2 · 2025-10-28

**Soundness:** 3
**Presentation:** 3
**Contribution:** 2
**Rating:** 4
**Confidence:** 3

**Summary:**

The paper investigates whether post-training alignment can enable large language models (LLMs) to transfer and generalize structured reasoning skills acquired from synthetic graph tasks to real-world problems with implicit graph structure. The authors instantiate alignment via two rule-based reward schemes and evaluate them across multiple model families and both synthetic and real-world datasets. They further analyze the limitations—most notably compositionality gaps from single-step to multi-step reasoning and reliability issues in intermediate reasoning—highlighting remaining challenges for robust generalization.

**Strengths:**

1. The manuscript is clearly written with a well-motivated problem statement and strong readability.
2. The authors conduct extensive experiments for the proposed post-training alignment, covering both synthetic and real-world settings, with relatively comprehensive analysis dimensions.
3. The paper offers an in-depth diagnosis of LLM bottlenecks for graph reasoning (e.g., compositionality gaps, multi-step hallucination), which is insightful for future work.

**Weaknesses:**

1. The methodological novelty primarily lies in the reward design and its application-oriented adaptation; overall, the work reads more like an engineering-focused instantiation of existing alignment algorithms (GRPO/DPO) to the graph-reasoning setting rather than a fundamentally new learning paradigm.
2. The experiments lack head-to-head comparisons with external SFT-based baselines (including representative methods cited in the introduction). Evaluations focus on intra-method stages (base/SFT/alignment), making it difficult to substantiate superiority over standard SFT.
3. From Table 2, on QWEN2.5-7B, direct SFT yields the best results; on LLaMA-3.1-8B, substantial gains appear only after prior SFT, and alignment still induces regressions on certain tasks—overall improvements are limited.
4. The narrative drifts in the second half: the paper pivots from “improving real-world performance via alignment” to “analyzing reasoning bottlenecks,” leading to a mismatch between stated goals and realized contributions.
5. Reported gains concentrate on a subset of tasks/configurations and lack consistency/robustness across benchmarks (including OOD scenarios), so the evidential support for the central claim remains relatively weak.

**Questions:**

1. In the introduction, the paper claims that existing SFT methods fail to learn generalizable reasoning patterns from structured reasoning data, leading to poor generalization on real-world tasks. The proposed post-training alignment is introduced as a remedy for this issue. However, the latter half of the paper shifts focus to analyzing LLM bottlenecks in graph reasoning. I am confused about how this analysis connects to your original motivation. If this shift occurs because the proposed alignment underperforms, I would suggest substantially revising the Introduction to clarify this transition and motivation.
2. The paper does not include horizontal comparisons with other methods, which raises concerns about the fairness and completeness of the evaluation.

---

### Note · Authors · 2025-11-15

**Comment:**

We would like to thank the reviewers for their thoughtful comments and feedback.

**Withdrawal Confirmation:**

I have read and agree with the venue's withdrawal policy on behalf of myself and my co-authors.